# Macronutrient Intake in Pregnancy and Child Cognitive and Behavioural Outcomes

**DOI:** 10.3390/children8050425

**Published:** 2021-05-20

**Authors:** Rachael M. Taylor, Michelle L. Blumfield, Lee M. Ashton, Alexis J. Hure, Roger Smith, Nick Buckley, Karen Drysdale, Clare E. Collins

**Affiliations:** 1Priority Research Centre in Physical Activity and Nutrition, University of Newcastle, Callaghan, NSW 2308, Australia; rachael.taylor@newcastle.edu.au (R.M.T.); lee.ashton@newcastle.edu.au (L.M.A.); 2Faculty of Health and Medicine, School of Health Sciences, University of Newcastle, Callaghan, NSW 2308, Australia; 3Department of Nutrition, Dietetics and Food, Faculty of Medicine, School of Clinical Sciences, Nursing and Health Sciences, Monash University, Clayton, VIC 3800, Australia; michelleblumfield@bigpond.com; 4Priority Research Centre for Generational Health and Ageing, University of Newcastle, Callaghan, NSW 2308, Australia; alexis.hure@newcastle.edu.au; 5Faculty of Health and Medicine, School of Medicine and Public Health, University of Newcastle, Callaghan, NSW 2308, Australia; roger.smith@newcastle.edu.au; 6Hunter Medical Research Institute, 1 Kookaburra Circuit, New Lambton Heights, NSW 2305, Australia; 7Priority Research Centre for Reproductive Science, University of Newcastle, Callaghan, NSW 2308, Australia; 8School of Psychology and Exercise Science, Murdoch University, Murdoch, WA 6150, Australia; nick.buckley@gmail.com; 9Faculty of Science, School Psychology, University of Newcastle, Callaghan, NSW 2308, Australia; karen.drysdale@newcastle.edu.au

**Keywords:** pregnancy, nutrition, cognition, behaviour, development, macronutrients

## Abstract

Prenatal nutrient exposures can impact on brain development and disease susceptibility across the lifespan. It is well established that maternal macronutrient intake during pregnancy influences foetal and infant development. Therefore, we hypothesise that macronutrient intakes during pregnancy are correlated with cognitive development during early childhood. The current study aimed to investigate the relationship between maternal macronutrient intake during pregnancy and child cognitive and behavioural outcomes at age 4 years. We analysed prospective data from a cohort of 64 Australian mother–child dyads. Maternal macronutrient intake was assessed using a validated 74-item food frequency questionnaire at 2 timepoints during pregnancy. Child cognition and behaviour were measured at age 4 years using the validated Wechsler Preschool and Primary Scale of Intelligence, 3rd version (WPPSI-III) and the Child Behaviour Checklist (CBC). Linear regression models were used to quantify statistical relationships and were adjusted for maternal age, education, pre-pregnancy BMI, breastfeeding duration and birthweight. Child Performance IQ was inversely associated with maternal starch intake (b = −11.02, *p* = 0.03). However, no other associations were found. Further research is needed to explore the association between different types of starch consumed during pregnancy and child cognitive development.

## 1. Introduction

The Developmental Origins of Health and Disease (DOHaD) hypothesis postulates that environmental exposures in utero or during the postnatal period may alter developmental physiology and susceptibility to disease across the lifespan [1]. Inadequate nutrition during pregnancy and the first 3 years of life may result in permanent functional changes in the brain and lead to psychopathology including schizophrenia [2] and antisocial externalizing behaviour, as this period is critical for brain growth [3,4]. Approximately 2 to 3 weeks after conception, foetal brain growth commences with the formation of the neural tube [5]. This is followed by significant neuronal proliferation, differentiation and migration during early gestation (8–22 weeks) [5]. Synaptogenesis, apoptosis and myelination are life-long neurological processes that commence during late gestation (24–35 weeks gestation) [5]. Adequate nutrients are necessary for the functioning of these neurodevelopmental processes [6].

The link between maternal nutrition and foetal brain development is clearly established from the analyses of offspring from the Dutch Winter Hunger famine of 1944–1945 [7,8]. These studies report that the risk of a psychiatric disorder during adulthood was higher in the offspring that were exposed to severe prenatal famine [7,8]. Emerging evidence also suggests that maternal overnutrition, an energy imbalance resulting in overweight and obesity, is associated with behavioural disorders in the offspring including attention deficit-hyperactivity disorder and autism spectrum disorder [9,10,11]. The functional changes in the brain caused by maternal nutritional insults maybe partially explained by epigenetic mechanisms including DNA methylation [12,13,14]. Studies conducted in the Gambia, a country in West Africa, indicate that seasonal fluctuations in maternal nutrient intake are correlated with DNA methylation patterns in the offspring at 2–8 months [15,16]. These findings suggest that adequate maternal nutrition is important for preventing aberrant DNA methylation patterns and phenotypes.

Maternal macronutrient intake is known to impact on foetal and infant growth [17,18,19,20]. Sloan et al. (18) reported that both low-protein (<50 g) and high-protein (≥85 g) during pregnancy has a quadratic (U-shaped) relation with foetal growth. Evidence suggests that impaired placental transport and umbilical uptake of amino acids may explain this relationship [21,22,23]. The Women and Their Children’s Health (WATCH) study, a prospective longitudinal birth cohort of 156 mother–child dyads, previously reported that maternal macronutrient profile was associated with foetal body composition [24]. Higher foetal abdominal fat was correlated with low maternal protein intake (<16% of total energy), while mid-thigh fat was highest at intermediate protein (18–21% of total energy), high fat (>40% of total energy) and low carbohydrate (<40% of total energy) intake [24]. While analysing the impact of maternal macronutrient intake on foetal and infant brain composition is not ethical in living human subjects, animal models have shown that protein restriction during pregnancy impairs the micro-structure of the foetal brain rather than altering the total brain weight [25]. A low protein intake during pregnancy adversely affects dendrite numbers [26], morphology of hippocampal cells [27], mossy fibre (MF) axonal area [28,29], synaptic spine complexity [27,30] and polyunsaturated fatty acid (PUFA) brain content [31,32]. Furthermore, animal studies have shown that maternal protein restriction is correlated with deficits in cognitive and behavioural function in offspring [30,33,34,35].

Currently, at a global and national scale, there is a lack of public health strategies for the prevention neurodevelopmental disorders [36]. Given the known link between maternal nutrition during pregnancy and future of health of the offspring [1], nutrition is one area that could be targeted. To date, human studies have primarily focused on the consequences of maternal micronutrient intake rather than macronutrient profile on child cognitive and behavioural function [37,38]. This is important as macronutrients (carbohydrates, protein and fat) are the main contributors to maternal and foetal energy intake, which is essential for foetal cellular and tissue growth [17,18,19]. In the human brain, glucose derived from dietary carbohydrates is the primary energy source; protein is required for the synthesis of neurotransmitters, enzymes and cell membranes; fat is abundant in the structural matrix of cell membranes, as well as myelin [39,40]. These roles and functions indicate that macronutrients are essential for optimal brain development and function. We hypothesise that maternal macronutrient profile during pregnancy is associated with child cognitive outcomes. This evidence will be important for informing dietary recommendations during pregnancy for supporting optimal child cognitive development and preventing neurodevelopmental disorders. Therefore, the aim of the current study was to investigate the relationship between maternal macronutrient intake during pregnancy on child cognitive and behavioural outcomes at age 4 years in an Australian population.

## 2. Materials and Methods

### 2.1. Study Population

The current study analysed prospective data from pregnant women and their children enrolled in The Women and Their Children’s Health (WATCH) study [41]. Pregnant women were recruited from the antenatal clinic at the John Hunter Hospital (JHH), New South Wales (NSW), Australia, from July 2006 to December 2008. Women were eligible for this study if they were: (1) less than 18 weeks pregnant, (2) lived in the local or neighbouring areas, and (3) were able to attend the JHH for study visits. Women were recruited by midwives, local media coverage, or by word of mouth. A consent rate of 61% was achieved for the pregnant women who were approached to participate in this study and 182 women were enrolled [42]. The recruitment, withdrawals and attendance of study participants have been described previously [43]. Women attended study visits during pregnancy at approximately 19, 24, 30 and 36 weeks gestation. Women and their children attended postnatal study visits at 3-monthly intervals during the first 12 months after birth, and then annually until the age of 4 years. Study visit attendance and participant withdrawals are reported in Figure 1. The WATCH study received ethics approval from the Hunter New England Research Ethics Committee (06/05/24/5.06) and therefore has been performed in accordance with the ethical standards laid down in the 1964 Declaration of Helsinki and its later amendments. All participants gave written informed consent prior to their inclusion in this study. The reporting of this study adheres to The Strengthening the Reporting of Observational Studies in Epidemiology (STROBE) Statement: guidelines for reporting observational studies [44].

### 2.2. Dietary Assessment

Dietary assessment methods used in the WATCH study have been described in detail elsewhere [24,41,45]. Briefly, dietary data during pregnancy were collected between 18 and 24 weeks and again between 36 and 40 weeks gestation. Dietary data were collected using the validated food frequency questionnaire (FFQ) and the Dietary Questionnaire for Epidemiological studies (DQES) version 2 [46]. The DQES has previously demonstrated an acceptable level of accuracy for estimating nutrient intake compared to 7 day weighed food records in women (*n* = 63) of child-bearing age [47]. The self-administered questionnaire required the women to use a 10-point frequency scale to report usual consumption of 74 foods (excluding vitamin and mineral supplements) and 6 alcoholic beverages over the previous 3 months. The 10-point frequency scale using categories ranged from never to three or more times per day. Photographs were used to represent different serving sizes for vegetables, potatoes and meat casserole dishes [46]. These photographs enabled the calculation of a portion factor to account for variations in serving size. The dietary data collected between 18 and 24 weeks and again between 36 and 40 weeks of gestation are referred to as reference periods of 6–24 weeks of gestation (early pregnancy) and 24–40 weeks of gestation (late pregnancy), respectively. The WATCH cohort has previously reported positive pairwise correlations between all dietary variables in early and late pregnancy (0.46 < *r* < 0.78; *p* < 0.001) [24]. Therefore, maternal dietary intakes during pregnancy were expressed as the mean of intakes during early and late pregnancy.

### 2.3. Dietary Analysis

The DQES is a computer-scannable questionnaire purchased at a price that includes dietary analysis by the FFQ distributor, Cancer Council Victoria. Nutrient intakes were quantified from the Nutrient Tables-1995 (NUTTAB-1995) database. Dietary analysis results were provided in a Microsoft Excel format to facilitate data importation for statistical analysis.

### 2.4. Cognition and Behavioural Assessment

#### 2.4.1. Cognition

Child cognition was assessed using the Wechsler Preschool and Primary Scale of Intelligence (WPPSI-III Australian) [48] which is suitable for children aged 4 to 7.3 years (PsychCorp, Sydney, Australia). The cognitive assessments were individually administered by a research psychologist at the 4 year study visit. The WPPSI-III is widely cited for preschool children and has satisfactory criterion validity, correlating with the Wechsler Preschool and Primary Scale of Intelligence, revised version (WPPSI-R), the Wechsler Intelligence Scale for Children, 3rd edition (WISC-III) and the Wechsler Intelligence Scale for Children, 4th edition (WISC-IV) [49,50]. The scale produces 3 main composite scores—the Full Scale Intelligence Quotient (FSIQ), the Performance Intelligence Quotient (PIQ) and the Verbal Intelligence Quotient (VIQ)—as well as 2 additional composite scores, the Processing Speed Quotient (PSQ) and the General Language Composite (GLC). The PIQ is derived from the scores of 5 subtests, which include Block Design, Matrix Reasoning, Picture Concepts, Picture Completion and Object Assembly. The VIQ is derived from scores of 5 subtests, which include information, vocabulary, word reasoning, comprehension and similarities. The PSQ is derived from the scores of 2 subtests— symbol search and coding. The GLC is derived from the scores of 2 subtests— receptive vocabulary and picture naming. The raw scale scores were converted to standardised scores according to the child’s age. The Full Scale IQ is the combined standardised scores derived from both the Performance IQ and Verbal IQ. All composite scores have a mean of 100 and a standard deviation of 15.

#### 2.4.2. Behaviour

Child behaviour was assessed using the Child Behaviour Checklist (CBC) for children aged 1.5 to 5 years [51], which has demonstrated internal accuracy of the scale across 22 countries, including Australia [52]. The behaviour assessments were completed by the primary caregiver of the child during their 4 year study visit. The checklist contains 113 behavioural/emotional problem items (questions) in 8 syndrome scales. The syndrome scales include anxious/depressed, withdrawn/depressed, somatic complaints, social problems, thought problems, attention problems, rule-breaking behaviour, and aggressive behaviour. The first 3 syndrome scales (anxious/depressed, withdrawn-depressed, and somatic complaints scores) combined to produce the internalizing problems score (internalizing broadband scale), and the last 2 syndrome scales (rule-breaking and aggressive behaviour) produce the externalizing problems score (externalizing broadband scale). The Total Behaviour Problem Scale summarises the scores obtained across all scale scores. The checklist items are rated by the child’s parent on a 3-point scale, ‘not true’ (0 point), ‘sometimes true’ (1 point) and ‘often true’ (2 points). Scores of the scales are interpreted as normal, borderline or clinical behaviour.

#### 2.4.3. Participant Characteristics

Sociodemographic, maternal, and medical data were collected from WATCH pregnant women during their first study visit, as previously reported [43].

### 2.5. Statistical Analysis

Maternal and child characteristics were summarised using median and interquartile range (IQR) for continuous variables, and frequencies and percentages for categorical variables. The maternal characteristics of the WATCH women (*n* = 69) included in this study were compared to women that were excluded due to missing dietary and/or cognition and behaviour data (*n* = 113) using Fisher’s exact tests. Further comparisons were made between the maternal age of both groups using a two-sample t-test. During early and late pregnancy, mean maternal intakes of energy, protein, total fat, saturated fatty acids (SFA), monounsaturated fatty acids, PUFA, omega-3 (*n* − 3) fatty acids, omega-6 (*n* − 6) fatty acids, total carbohydrate, total sugars (including fructose, glucose, sucrose, maltose, lactose and galactose), starch (polysaccharides including amylose, amylopectin, glycogen and dextrins) and fibre (souble and non-souble sources including non-starch polysaccharides and lignin) were expressed as kilojoules (KJ), total intakes (grams), percentage of total energy intake (%E), and the ratio of protein to carbohydrate (P:C) and ratio of fatty acids *n* − 6:*n* − 3. Nutrient intakes were adjusted for energy using the residual method [53]. Child body mass index (BMI) was converted to z-scores using the World Health Organization (WHO) Child Growth Standard references data [54]. To address potential misreporting of dietary intake, the pregnancy energy cut-off values recommended by Meltzer et al. [55] were applied which excluded women who reported daily energy intakes <4.5 or >20.0 MJ/day. Robust linear regression models were developed to determine the association of maternal macronutrient intake on child cognitive and behavioural outcomes. Maternal macronutrient intake data were transformed (natural logarithm) to achieve linearity. Analyses were adjusted for maternal age, education, pre-pregnancy BMI, birthweight and duration of breastfeeding (weeks) as they are significant predictors for child cognitive and behavioural outcomes. Sensitivity analyses were conducted using linear regression models that were not adjusted for total energy intake nor confounders to assess the impact on the study results. All tests assumed a 5% significance level. All statistical analyses were performed using STATA 13 (Stata, College Station, TX, USA).

## 3. Results

Of the 182 women recruited, 69 women provided dietary data during early (6–24 weeks gestation) and late pregnancy (24–40 weeks gestation) (Figure 1). Five women reported implausible dietary intakes and were excluded from the analysis. Maternal dietary data and child cognition data were available from 58 mother–child dyads, while maternal dietary data and child behavioural data were available from 51 mother–child dyads. The characteristics of the WATCH mother–child dyads are summarised in Table 1. In summary, the characteristics of the women were median (IQR) 29 (7) years of age, married (63%), with 36% having attained a university degree and 89% were non-smokers. Birthweights were median (IQR) 3590 (770) grams.

Sociodemographic, maternal, and medical data were self-reported by the participants in a questionnaire.

Cognition scores at age 4 years were: Full Scale IQ 108 (99–114), Verbal IQ 105 (98–111), Performance IQ 107 (100–118), PSQ 108 (101–114) and GLC 108 (97–117). For Full Scale IQ 66% of the WATCH children exceeded the mean ± SD of Australian norms (100 ± 15) [51]. The median(IQR) behaviour scores at age 4 years were: emotionally reactive 54 (50–81), anxious/depressed 54 (50–73), somatic complaints 50 (50–62), withdrawn 73 (54–84), sleep problems 54 (50–73), attention problems 54 (50–76), aggressive behaviour 50 (50–73), and stress 62 (54–90), total problems 50 (17–69), internalizing 46 (24–79) and externalizing 38 (16–76). For total problems, internalizing and externalizing behaviour, 38%, 39% and 33% of the WATCH children exceeded the American norms (50 ± 10), respectively [51].

The maternal macronutrient composition of the dietary intakes of the pregnant women are summarised in Table 2. Carbohydrate intake (median: 42% of total energy intake) of the WATCH cohort was within the acceptable macronutrient distribution range (AMDR) of 45–65% of total energy intake [56]. Median starch intake was 99.2 g per day. Currently, there is no nutrient reference value (NRV) set for starch intake and, therefore, low, medium and high consumers could not be determined. Protein intake (median: 19% of total energy intake) was within the AMDR of 5–20% of total energy intake [56]. Total fat intake (median: 37% of total energy intake) of the WATCH women was above the AMDR of 20–35% of total energy intake [56]. Maternal saturated fat intake (median: 16% of total energy intake) exceeded the AMDR of ≤10% of total energy intake, consistent with other studies [57,58,59].

Results of the mixed-models regression analyses examining the association between macronutrient composition during pregnancy and child cognitive and behavioural outcomes are provided in Table 3 and Table 4. There was a non-significant trend indicated for each log-transformed additional gram of total carbohydrate intake consumed during pregnancy, child Performance IQ decreased (worsened) by approximately 15% (25 points out of a maximum score of 160). The log-transformed maternal starch intake during pregnancy was negatively associated with performance IQ (*b* = −11.02, *p* = 0.03). For each log-transformed additional gram of starch consumed during pregnancy, child performance IQ was lower by approximately 11 points out of a maximum score of 160. Based on the R-squared value, 26% of the variation in the outcome was accounted for in the linear regression model. Although this association was not statistically significant when maternal starch intake was not adjusted for energy intake (*b* = −8.06, *p* = 0.07 (Appendix A)). We have estimated that the Australian Guide to Healthy Eating (AGHE) [60] core food groups (i.e., bread and cereals, vegetables, fruit, dairy and alternatives, meat and alternatives) and the non-core foods (i.e., energy-dense nutrient poor foods that provide 600 KJ per serve) contributed to 76% and 24% of maternal starch intake per day (data not shown).

Child externalizing behaviour did show a positive trend with log-transformed maternal protein (*b* = 27.71, *p* = 0.05) and sugar intake (*b* = 24.19, *p* = 0.05) but was not statistically significant. However, when the data were adjusted for breastfeeding duration (weeks), the strength of these associations was reduced (*p* ≥ 0.05). The associations between crude maternal nutrient intake and child behavioural outcomes are provided in Appendix A. Analysis models without adjustments for confounding factors are provided in Appendix A. These sensitivity analyses did not detect any significant differences compared to the study findings presented. Maternal P:C ratio was not associated with child cognitive or behavioural outcomes, *p* > 0.05. The R-squared values for the linear regression models indicated that each adjusted model accounted for between 5 to 32% of the total variance in the child cognitive or behavioural outcomes.

## 4. Discussion

To our knowledge, this is the first human study that has evaluated the association between macronutrient profiles during pregnancy on subsequent child cognitive and behavioural outcomes at age 4 years. This study identified increasing carbohydrate intake during pregnancy was negatively associated with child Performance IQ, a measure of non-verbal reasoning, attention and visuo-spatial processing. However, this relationship was not significant. Although there is evidence to support that an abnormal carbohydrate metabolism during pregnancy such as impaired glucose tolerance (IGT) and gestational diabetes mellitus (GDM) is adversely associated with child cognitive outcomes [61,62,63]. For example, Xu et al. [62] reported that children born to mothers with GDM (*n* = 1421) had lower total wide-range assessment of visual motor abilities scores (WRAVMA), a measure of visual-spatial and fine motor ability, at 3 years of age compared with children born to mothers with normal glucose tolerance (*n* = 1187) (−3.09 points; 95% confidence interval (CI) −6.12, −0.05). Interestingly, child Performance IQ was negatively associated with maternal starch intake during pregnancy in which the relationship was statistically significant (*p* = 0.03). Starch is a complex carbohydrate that consists of polysaccharides, consumed from plant-based foods (e.g., oats, rice, potato) [64]. In the WATCH cohort, maternal starch intake (median: 99.2 g per day) only varied by 2% compared to the 2011–2012 national averages reported in women 19–50 years [65]. The current study reported that non-core foods significantly contributed (24%) to maternal starch intake [65]. These findings are supported by national data which indicated that processed cereal products and dishes (e.g., biscuits, pastries, pizza) are a major contributor (31%) to daily starch intake in women aged 19–50 years [65]. This is concerning as these foods contain a higher proportion of rapidly digested starches and significantly less resistant starches which are lost during various food processing methods (e.g., milling, cooking, high-pressure processing) [66,67]. A higher consumption of rapidly digested starches contributes to increased de novo lipogenesis and attenuates the deposition of triglycerides into adipocytes throughout the body [68]. Triglycerides are chemically stable when stored in adipocytes [69,70,71]. However, once storage is at saturation, triglycerides may be deposited in non-adipose tissues such as the liver, heart and pancreas, which can lead to lipotoxicity and inflammation [69,70,71]. In the current study, it could be hypothesised that higher intakes of rapidly digested starches alters lipid metabolism and adversely impacts on cell, tissue and organ structure and function in the central nervous system, potentially contributing to a decline in child Performance IQ. This could not be evaluated in the current study, as specific types of starch could not be quantified using NUTTAB-95 data. Therefore, further investigation is warranted in future studies where detailed information on types of starch is available.

Resistant starch is the portion of starch that is resistant to degradation by the enzyme α-amylase in the small intestine [72]. Instead, it is fermented in the colon by several bacteria groups (e.g., amylolytic gut bacteria) releasing fermentation products including short-chain fatty acids (acetate, propionate, butyrate, and valerate), branched-chain fatty acids (isovaleric and isobutyric acids), ammonia, amines, phenolic compounds and gases (methane, hydrogen, carbon dioxide) [72]. Colonic metabolites including short-chain fatty acids (acetate, propionate, butyrate, and valerate) are associated with a number of health benefits on gastrointestinal health, insulin sensitivity and weight management [72,73]. For example, short-chain fatty acids propionate stimulates the secretion of gut hormone peptides YY (PYY) and glucagon-like peptide 1 (GLP-1) which are essential for appetite regulation and glucose homeostasis [74]. Butyrate is associated with being an anti-inflammatory agent by inhibiting the activation of transcription factors, NF-kB which regulates the expression of genes associated with inflammation (e.g., cytokines, adhesion molecules, acute-phase proteins) [75].

The impact of resistant starch on brain function and cognition has not been well explored [76,77,78]. However, emerging evidence suggests that resistant starch may impact on cognitive function by altering the serotonergic (5-HT) system that controls the activity of neurotransmitters [79,80]. Animal studies have shown that a high fat diet in rats during adulthood alters the density of serotonergic receptors in the brain [81,82] and such adverse changes can be reversed by the intake of resistant starch and galacto-oligosaccharides [76]. Further research is greatly needed in this area, especially during pregnancy, to provide further insight in relation to these findings. This evidence will be important as modifying resistant starch intake could potentially be used as a therapeutic intervention for improving brain function and cognition.

Limitations in the current study need to be acknowledged and include the use self-reported dietary data. Therefore, the possibility of mis-reporting cannot be excluded. To improve the validity of reported total energy intakes, the pregnancy energy cut-off values recommended by Meltzer et al. [55] were applied (<4.5 or >20.0 MJ/day). Dietary data are further strengthened by similarities between the WATCH median energy-adjusted intake (7317 KJ) and the Australian Longitudinal Study on Women’s Health (ALSWH) mean energy intake (7795 KJ) [83]. Maternal macronutrient profile was similar to the 2011–2012 national averages reported in women aged 19–50 years [65]. The most variation can be seen in the median maternal fat intake, which was 5% higher compared to the national average [65]. While median protein and carbohydrate intake only varied by 1% compared to the national average [65]. These findings indicate that the dietary intake of this cohort is generalizable to women in the Australian population. The WATCH study has previously demonstrated a positive correlation between all dietary variables during early and late pregnancy [24]. Maternal nutrient intake was quantified using NUTTAB-1995 which was the most comprehensive database at the time of the study. However, specific data were not available for new food products produced after 1995. While study findings may be confounded by postnatal diet which was not explored in the current analyses. Although the regression models were adjusted for breastfeeding duration. Previously analyses from the WATCH cohort demonstrated that maternal diet during 2–3 years of the postnatal period was correlated with the overall dietary quality in the offspring (*p* < 0.001) [84].

The measurement of cognition and behaviour are known to be affected by a child’s mood, motivation, anxiety, energy levels, and personal effort [85]. To address this, the WATCH study monitored the children’s energy levels and mental concentration and prioritized the cognition subsets that were necessary to produce IQ scores. Furthermore, child cognitive function is also influenced by genetics, biomedical, social and environmental factors that have not been explored in the current analyses [86,87]. Child cognitive and behavioural outcomes were not deemed the primary outcomes when the WATCH study was established. Due to the extended length of follow up, attrition was high and the sample size was small (*n* = 64). This study is likely to be underpowered to detect the associations between maternal macronutrient intake and child cognition and behavioural outcomes. The analysis of this study was exploratory and therefore multiple variables were used to provide a wide scope of data and inform future research in this area. However, the use of multiple predictors within a small sample size may have increased the risk of type I error within the dataset. Lastly, causality cannot be inferred as this is a prospective cohort study. Therefore, the observed relationships require further investigation in experimental studies.

## 5. Conclusions

This prospective cohort study found that child Performance IQ at 4 years was inversely associated with maternal starch intake. There is an opportunity for future cohort studies to investigate the relationship between intake of different types of starch consumed during pregnancy with lipid metabolism and child cognitive development.

## Figures and Tables

**Figure 1 children-08-00425-f001:**
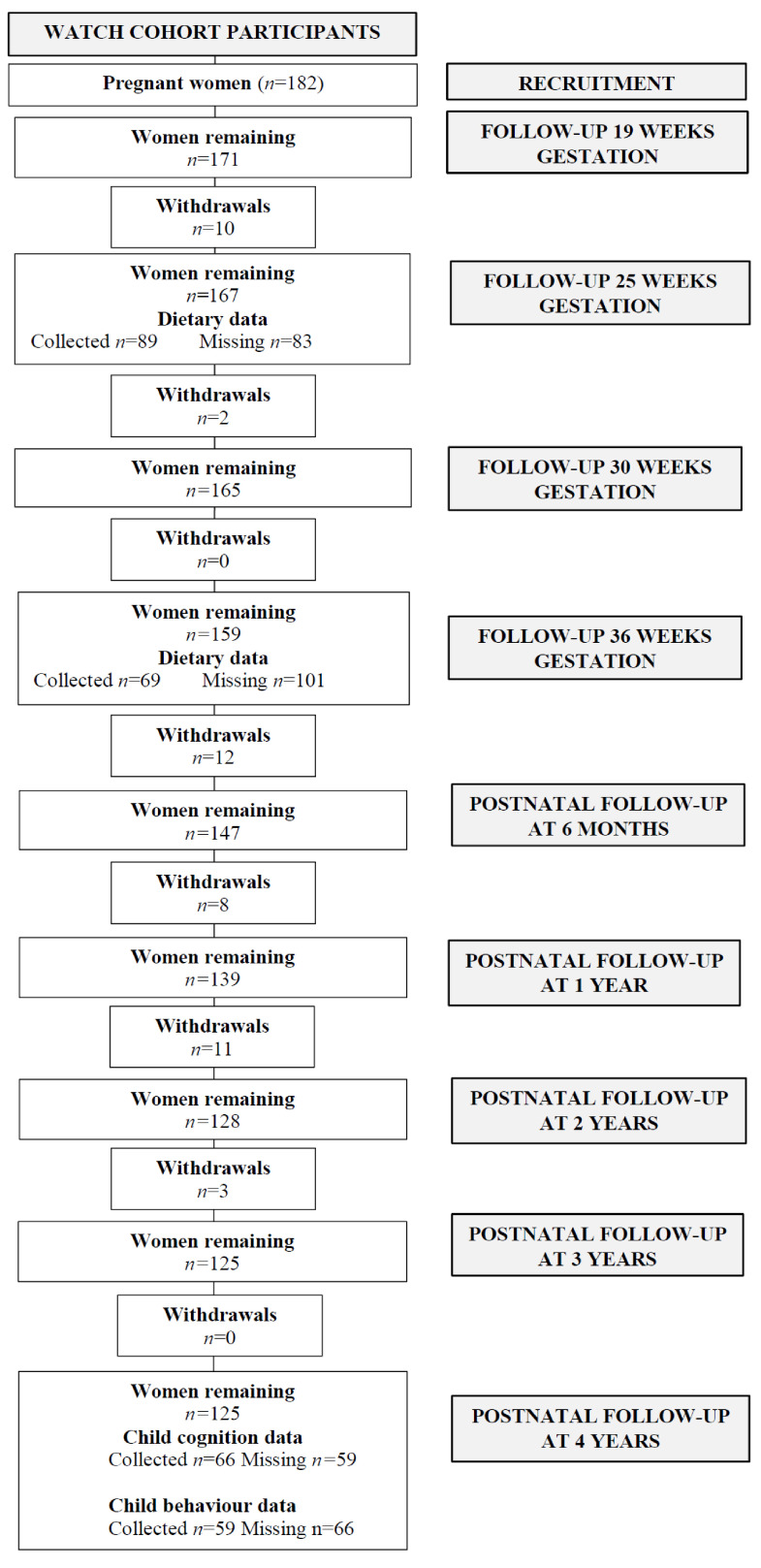
Flowchart of mother–child pairs enrolled in the WATCH cohort and included in the statistical analysis.

**Table 1 children-08-00425-t001:** Characteristics of the WATCH mother–child dyads included in the analysis (*n* = 64).

Characteristics		
Pregnant Women	Median (IQR) ^1^	Range Difference
Maternal Age (y)	29 (7)	22.4
Education	***n***	**%**
No formal qualification	1	1.6
Year 10 or equivalent	10	16
Year 12 or equivalent	11	17
Trade/apprenticeship	2	3.1
Certificate/diploma	14	22
University degree	23	36
Higher university degree	3	4.7
Missing	0	0
Household Weekly Income	***n***	**%**
No income	0	0
$AUD ^1^ 1–299	4	6
$AUD ^1^ 300–699	13	20
$AUD ^1^ 700–999	13	20
$AUD ^1^ 1000 or more	30	47
Unsure	4	6
Missing	0	0
Marital Status	***n***	**%**
Never married	20	32
Married	40	63
Separated/divorced	3	4.8
Widowed	0	0
Missing	1	1.6
Maternal Smoking	***n***	**%**
Yes	7	11
No	57	89
Missing	0	0
Maternal Depression	***n***	**%**
Yes	17	27
No	46	72
Missing	1	1.6
Maternal Anxiety	***n***	**%**
Yes	9	14
No	54	84
Missing	1	1.6
Previous Live Births(>37 Weeks Gestation)	***n***	**%**
None	34	53
1–2	26	41
3–4	4	6.2
>5	0	0
Missing	0	0

^1^ $AUD, Australian dollars; IQR, interquartile range.

**Table 2 children-08-00425-t002:** Maternal dietary composition * during pregnancy (*n* = 69).

Nutrients	Daily NRVs	Daily Intake	% of Energy
Energy (KJ)		7095.7 (5860.3, 8610.73)	*n*/a
Protein (g)	5–20 ^†^	78.5 (66.0, 101.7)	19.4 (17.8, 21.1)
Total fat (g)	20–35 ^†^	71.2 (58.1, 85.9)	37.3 (34.2, 39.7)
SFA (g)	≤10 ^†^	29.4 (23.2, 36.9)	15.9 (13.1, 17.7)
PUFA (g)		10.9 (8.1, 13.0)	5.2 (4.4, 6.5)
MUFA (g)		24.5 (19.9, 29.4)	12.7 (11.8, 13.9)
Total carb. (g)	45–65 ^†^	181.8 (153.7, 234.3)	42.1 (39.5, 44.8)
Sugars (g)	≤25% ^†^	89.9 (72.3, 110.6)	19.9 (17.3, 22.0)
Starch (g)		96.9 (79.6, 118.2)	21.3 (20.0, 23.6)
Fibre (g)	25 (AI) ^§^	19.4 (15.1, 23.7)	4.3 (3.7, 4.9)
P:C ratio (g)		2.0 (1.0, 3.0)	*n*/a
Energy-adjusted values (*n =* 64)			
Energy (KJ)		7317.0 (5984.2, 8706.3)	*n*/a
Protein (g)	5–20 ^†^	81.1 (69.2, 103.6)	19.2 (17.7, 21.0)
Total fat (g)	20–35 ^†^	72.3 (60.4, 87.4)	37.3 (34.7, 40.0)
SFA (g)	≤10 ^†^	30.3 (25.0, 38.8)	16.0 (13.4, 17.8)
PUFA (g)		11.2 (8.8, 13.1)	5.2 (4.4, 6.6)
MUFA (g)		24.9 (20.9, 31.4)	12.6 (11.8, 13.9)
Total carb. (g)	45–65 ^†^	186.1 (156.8, 237.6)	42.0 (39.5, 44.7)
Total Sugars (g)	≤25% ^†^	92.6 (74.3, 113.2)	19.9 (16.6, 22.0)
Starch (g)		99.2 (80.4, 120.9)	21.3 (19.8, 23.7)
Fibre (g)	25 (AI) ^§^	20.9 (15.7, 24.3)	4.2 (3.6, 4.9)
P:C ratio (g)		2.0 (1.0, 3.0)	*n*/a

* All values are medians (25th and 75th percentiles). AI, adequate intake; carb., carbohydrate; P:C, protein-to-carbohydrate; SFA, saturated fatty acids; PUFA, polyunsaturated fatty acids; MUFA, monounsaturated fatty acids; NRVs, nutrient reference values. All values are medians (25th and 75th percentiles). ^†^ Food and Nutrition Board: Institute of Medicine (FNB: IOM) reference values used for adults—carbohydrates: 45–65% energy intake, added sugar: ≤25% of total energy intake, protein: 5–20% energy intake, total fat: 20–35% energy intake, and saturated fatty acids: ≤10% energy intake. ^§^ National Health and Medical Research Council (NHMRC) Nutrient Reference Values for women aged 19–50 years.

**Table 3 children-08-00425-t003:** Association of maternal dietary composition during pregnancy with child cognition outcomes up to age 4 years (*n* = 58).

Variables ^1^	Beta- Coefficient	95% Confidence Interval	*p*-Value ^2^	*R*-Value
**Full Scale IQ**				
Energy	−1.27	−11.45 to 8.91	0.80	0.15
Protein (% E)	5.29	−19.53 to 30.10	0.68	0.16
Total fat (% E)	14.01	−9.19 to 37.20	0.23	0.18
PUFA (% E)	1.70	−6.96 to 10.37	0.70	0.15
CHO (% E)	−20.55	−46.12 to 5.01	0.11	0.19
P:C ratio	−0.73	−7.20 to 5.75	0.82	0.15
Protein (g)	0.33	−9.79 to 9.13	0.94	0.15
PUFA (g)	0.53	−6.61 to 7.67	0.88	0.05
Total sugars (g)	−0.51	−8.50 to 7.49	0.90	0.15
Starch (g)	−6.81	−16.02 to 2.40	0.14	0.19
**Verbal IQ**				
Energy	3.79	−9.73 to 17.31	0.58	0.06
Protein (% E)	−1.51	−34.61 to 31.59	0.93	0.05
Total fat (% E)	19.83	−10.80 to 50.45	0.20	0.08
PUFA (% E)	0.20	−11.21 to 11.61	0.97	0.05
CHO (% E)	−19.41	−53.73 to 14.91	0.26	0.07
P:C ratio	−3.56	−12.12 to 5.00	0.41	0.06
Protein (g)	3.05	−9.51 to 15.62	0.63	0.05
PUFA (g)	1.98	−7.45 to 11.41	0.68	0.05
Total sugars (g)	2.46	−8.16 to 13.09	0.64	0.05
Starch (g)	−2.23	−14.75 to 10.27	0.72	0.05
**Performance IQ**				
Energy	−5.75	−17.14 to 5.64	0.32	0.20
Protein (% E)	14.14	−13.68 to 41.96	0.31	0.20
Total fat (% E)	11.70	−14.69 to 38.11	0.38	0.20
PUFA (% E)	−0.39	−10.19 to 9.42	0.94	0.19
CHO (% E)	−24.67	−53.48 to 4.14	0.09	0.23
P:C ratio	1.85	−5.46 to 9.15	0.61	0.19
Protein (g)	−2.91	−13.56 to 7.74	0.59	0.19
PUFA (g)	−3.09	−11.12 to 4.93	0.44	0.20
Total sugars (g)	−3.78	−12.71 to 5.16	0.40	0.20
Starch (g)	−11.02	−21.19 to −0.84	0.03	0.26
**Processing Speed Composite**				
Energy	0.92	−9.83 to 11.67	0.86	0.18
Protein (% E)	2.73	−20.56 to 26.03	0.81	0.18
Total fat (% E)	6.14	−16.24 to 28.52	0.58	0.19
PUFA (% E)	−0.13	−8.09 to 7.84	0.98	0.18
CHO (% E)	−12.23	−36.63 to 12.17	0.32	0.20
P:C ratio	3.17	−2.72 to 9.05	0.28	0.20
PUFA (g)	0.31	−6.90 to 7.52	0.93	0.18
Protein (g)	1.14	−8.22 to 10.49	0.81	0.18
Total sugars (g)	0.80	−6.86 to 8.47	0.83	0.18
Starch (g)	−4.69	−15.57 to 6.19	0.39	0.20
**General Language Composite**				
Energy	2.86	−12.76 to 18.49	0.72	0.06
Protein (% E)	4.05	−34.13 to 42.24	0.83	0.05
Total fat (% E)	22.65	−12.72 to 58.01	0.20	0.09
PUFA (% E)	4.01	−9.11 to 17.14	0.54	0.06
CHO (% E)	−24.03	−63.56 to 15.51	0.23	0.08
P:C ratio	−1.88	−11.82 to 8.05	0.71	0.06
Protein (g)	3.05	−11.46 to 17.56	0.68	0.06
PUFA (g)	4.14	−6.71 to 14.98	0.45	0.07
Total sugars (g)	3.04	−9.21 to 15.30	0.62	0.06
Starch (g)	−6.11	−20.47 to 8.24	0.40	0.07

CHO, carbohydrates; P:C, protein to carbohydrate; PUFA, polyunsaturated fatty acids; % E, percentage of energy. Analysis models were adjusted for energy intake, maternal age, education, pre-pregnancy BMI, breastfeeding duration (weeks) and birthweight. ^1^ The natural logarithm transformation of the nutrient variable was used for the linear regression models to meet normality assumptions. ^2^ *p*-values were derived by linear regression models.

**Table 4 children-08-00425-t004:** Association of maternal dietary composition during pregnancy with child behaviour outcomes up to age 4 years (*n* = 51).

Variables ^1^	Beta- Coefficient	95% Confidence Interval	*p*-Value ^2^	*R*-Value
**Total Problems Score**				
Energy	15.02	−15.61 to 45.66	0.33	0.25
Protein (% E)	18.11	−56.01 to 92.22	0.63	0.23
PUFA (% E)	−19.98	−47.31 to 7.34	0.15	0.27
Total fat (% E)	−14.87	−90.95 to 61.21	0.70	0.13
CHO (% E)	26.5887	−57.25 to 110.42	0.53	0.24
P:C ratio	12.88	−6.93 to 32.69	0.20	0.26
Protein (g)	15.55	−12.71 to 43.81	0.27	0.25
PUFA (g)	−4.60	−26.18 to 16.98	0.67	0.23
Total sugars (g)	17.51	−6.50 to 41.51	0.15	0.27
Starch (g)	12.0573	−18.14 to 42.2	0.43	0.24
**Internalizing Broad Band Score**				
Energy	−1.47	−32.06 to 29.13	0.92	0.27
Protein (% E)	4.11	−69.30 to 77.52	0.91	0.27
Total fat (% E)	−10.67	−85.89 to 64.54	0.78	0.27
PUFA (% E)	−21.99	−48.82 to 4.84	0.11	0.32
CHO (% E)	23.54	−59.35 to 106.44	0.57	0.28
P:C ratio	11.54	−8.10 to 31.18	0.24	0.30
Protein (g)	−0.64	−28.95 to 27.66	0.96	0.27
PUFA (g)	−13.84	−34.79 to 7.11	0.19	0.30
Total sugars (g)	4.56	−19.69 to 28.81	0.71	0.27
Starch (g)	−0.26	−30.30 to 29.79	0.99	0.27
**Externalizing Broad Band Score**				
Energy	23.00	−7.57 to 53.58	0.14	0.22
Protein (% E)	18.93	−56.11 to 93.97	0.61	0.19
Total fat (% E)	1.68	−75.49 to 78.86	0.97	0.18
PUFA (% E)	−17.55	−45.39 to 10.29	0.21	0.21
CHO (% E)	−0.52	−88.14 to 87.11	0.99	0.11
P:C ratio	11.48	−8.67 to 31.63	0.26	0.21
Protein (g)	22.50	−5.70 to 50.70	0.12	0.23

CHO, carbohydrates; P:C, protein to carbohydrate; PUFA, polyunsaturated fatty acids; % E, percentage of energy. Analysis models were adjusted for energy intake, maternal age, education, pre-pregnancy BMI, birthweight and breastfeeding duration (weeks). ^1^ The natural logarithm transformation of the nutrient variable was used for the linear regression models to meet normality assumptions. ^2^ *p*-values were derived by linear regression models.

## Data Availability

Data are contained within the article or supplementary material.

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
