# Peer review of "Macronutrient Intake in Pregnancy and Child Cognitive and Behavioural Outcomes"

_children, 2021, doi:10.3390/children8050425_

Round 1
Reviewer 1 Report
This is a thorough study of gestational nutrition and cognitive functioning at age 4. The study is thorough and the results are clear. To this reader, a clear conclusion is that there is surprisingly little relationship between nutrition and cognitive outcome at 4 years of age. This is not a conclusion that the authors make, however.
There is one statistically significant effect - Performance IQ and starch intake - but given the large number of statistical tests and no apparent correction for this, this one finding is not compelling to this reader.
The authors correctly note that they may be under powered, which may account for their lack of a relationship between nutrition and cognitive outcome.
I also wonder if participant factors such as education, income, and marital status might be co-factors to consider. The authors collected these data but do not seem to use it in their analyses.
I puzzled by the absence to look at the association between Verbal IQ and maternal dietary composition. They show FS IQ and P IQ and "Nonverbal IQ" but do not explain what the latter measure is. Could this be a typo and it should be "Verbal IQ"?
Author Response
Reviewer 1
This is a thorough study of gestational nutrition and cognitive functioning at age 4. The study is thorough and the results are clear. To this reader, a clear conclusion is that there is surprisingly little relationship between nutrition and cognitive outcome at 4 years of age. This is not a conclusion that the authors make, however. There is one statistically significant effect - Performance IQ and starch intake - but given the large number of statistical tests and no apparent correction for this, this one finding is not compelling to this reader. The authors correctly note that they may be under powered, which may account for their lack of a relationship between nutrition and cognitive outcome. I also wonder if participant factors such as education, income, and marital status might be co-factors to consider. The authors collected these data but do not seem to use it in their analyses.
Author’s response:
All statistical models were adjusted for energy intake, maternal age, education, pre-pregnancy BMI, birthweight and breastfeeding duration (weeks). Maternal education is associated with
Income [1] and marital status were not found to be significant predictors of child cognition in the presented study.
I puzzled by the absence to look at the association between Verbal IQ and maternal dietary composition. They show FS IQ and P IQ and "Nonverbal IQ" but do not explain what the latter measure is. Could this be a typo and it should be "Verbal IQ"?
Author’s response:
Our apologies this is a typo and it should be Verbal IQ we have now made this correction.
Reference
- Library O. Indicator A4. What are the earnings advantages from education? Paris, France OECD Publishing; 2019
Reviewer 2 Report
Taylor et al. report a paper entitled ‘Macronutrient intake in pregnancy and child cognitive and behavioural outcomes.’ This study is soundly carried out and of interest to the literature, in my opinion.
- The Abstract is generally sound but no hypothesis is mentioned. Use of an hypothesis would give the study a clearer focus. ‘Exploration’ is a bit vague. What are you looking for? Why? Why might it be important? Thorough answers to these questions are only really appropriate for the Introduction but a clear focus is useful.
- The clear result of carbohydrate being associated with childhood IQ is of interest, however the discussion of this result is rather unedifying. There is barely any mention at all of nutritional programming, which is likely to be very important. Dysregulations of carbohydrate metabolism (such as hyperglycaemia) in pregnancy are not mentioned anywhere in the manuscript. Increased carbohydrate intake leads to higher rate of de novo lipogenesis, and thus has an important impact on lipid metabolism which is itself important in CNS development, which is not mentioned at all. This means that all of the principal mechanisms that we might expect to be involved in connecting maternal macronutrient intake and the functional behaviour of children are not discussed.
The programming effects of high carbohydrate and high fat diets by both mothers and fathers are becoming better understood, however none of these studies are cited. A recent paper has shown that the difference between a normal and a high carbohydrate intake by fathers has an important impact on the lipid metabolism in two generations of offspring and thus one would expect this to be magnified in mothers. Much more mechanistic reasoning is required, otherwise the study is merely ‘ooh look, carbohydrate might be important’. The thoroughness of your measurements deserve more than that.
- We are told that the results are corrected for various confounding factors. However, it is not clear what effect this has had. This should be made clear as associations cannot reasonably be made only on the basis of adjusted data. I suggest a sensitivity analysis to make this clear.
- The English and presentation could do with a check. “4 years” does not require a hyphen, neither does inform (line 37).
I look forward to reading a revised manuscript.
Author Response
The Abstract is generally sound but no hypothesis is mentioned. Use of an hypothesis would give the study a clearer focus. ‘Exploration’ is a bit vague. What are you looking for? Why? Why might it be important? Thorough answers to these questions are only really appropriate for the Introduction but a clear focus is useful.
Author’s response:
We can not include further information in the abstract as the journal specifies a word limit of 200 words. However, we have now provided a hypothesis and attempted to answer the reviewers questions in the introduction section. Lines 98-101 now reads:
“We hypothesise that maternal macronutrient profile during pregnancy is associated with child cognitive outcomes. This evidence will be important for informing dietary recommendations during pregnancy for supporting optimal child cognitive development and preventing neurodevelopmental disorders.”
The clear result of carbohydrate being associated with childhood IQ is of interest, however the discussion of this result is rather unedifying. There is barely any mention at all of nutritional programming, which is likely to be very important. Dysregulations of carbohydrate metabolism (such as hyperglycaemia) in pregnancy are not mentioned anywhere in the manuscript. Increased carbohydrate intake leads to higher rate of de novo lipogenesis, and thus has an important impact on lipid metabolism which is itself important in CNS development, which is not mentioned at all. This means that all of the principal mechanisms that we might expect to be involved in connecting maternal macronutrient intake and the functional behaviour of children are not discussed. The programming effects of high carbohydrate and high fat diets by both mothers and fathers are becoming better understood, however none of these studies are cited. A recent paper has shown that the difference between a normal and a high carbohydrate intake by fathers has an important impact on the lipid metabolism in two generations of offspring and thus one would expect this to be magnified in mothers. Much more mechanistic reasoning is required, otherwise the study is merely ‘ooh look, carbohydrate might be important’. The thoroughness of your measurements deserve more than that.
Author’s response:
Thank you for your thoughtful comment. In this study we did not find an association between overall maternal carbohydrate intake and child cognitive and behavioural outcomes. An association was found between maternal starch intake, a particular type of carbohydrate that includes polysaccharides, and child Performance IQ. Although, there seemed to be a non-significant (p=0.09) trend that higher intakes of total carbohydrates during pregnancy were adversely associated with child Performance IQ (beta-coefficient, 95%CI: –24.67, -53.48 to 4.14). We agree that a dysfunction in the metabolism of carbohydrates can adversely impact brain function however, based on the current evidence it is unclear how specifically starch intake may impact on this relationship. We have now acknowledged this in the results and discussion sections, lines 268-271 now reads:
There was a non-significant trend for each log transformed additional gram of total carbohydrate intake consumed during pregnancy, child Performance IQ was decreased by approximately 25 points out of a maximum score of 160.
Lines 309-313 now reads:
This study identified increasing carbohydrate intake during pregnancy was negatively associated with child Performance IQ however, this relationship was not significant. Although experimental evidence indicates that a dysfunction in carbohydrate metabolism may have detrimental effects on the developing brain and cognitive function.
Lines 326-328 now reads:
A higher consumption of rapidly digested starches contributes to a significant rise in blood glucose levels which can cause cell, tissue or organ injury and result in the development of chronic diseases [66-68].
Lines 334-336 now reads:
Resistant starch is the portion of starch that can not be digested in the small intestine but is fermented in the colon, producing metabolites, including short-chain fatty acids and provides a slow and prolonged release of glucose in the bloodstream [69].
We are told that the results are corrected for various confounding factors. However, it is not clear what effect this has had. This should be made clear as associations cannot reasonably be made only on the basis of adjusted data. I suggest a sensitivity analysis to make this clear.
Author’s response:
We have conducted a sensitivity analysis using data without adjustments for confounding factors and did not find any significant differences compared to the study findings presented. The sensitivity analysis is now provided in Supplementary Tables 3-4 which is indicated in the text of the manuscript.
The English and presentation could do with a check. “4 years” does not require a hyphen, neither does inform (line 37).
Author’s response:
We have now made these corrections in the manuscript.
Round 2
Reviewer 1 Report
The authors have clarified points that I raised although I am still unsure that their conclusions are justified by the nonsignificant results.
Author Response
Reviewer 1
The authors have clarified points that I raised although I am still unsure that their conclusions are justified by the non-significant results.
Author’s response:
We have now revised the conclusion so it is more consistent with the discussion section of the manuscript, lines 424-426 read:
‘This prospective cohort study found that child Performance IQ at 4-years was inversely associated with maternal starch intake. There is an opportunity for future cohort studies to investigate the relationship between intake of different types of starch consumed during pregnancy with lipid metabolism and child cognitive development.’
Reviewer 2 Report
Taylor et al. Present a revised manuscript ‘Macronutrient intake in pregnancy and child cognitive and behavioural outcomes’. The authors have made some of the revisions, but it is not clear that all of them have been implemented thoroughly enough.
- The Abstract.
This has not really been changed—the hypothesis needs to be stated in the Abstract. This can be done in one sentence, after brief indication of the evidence that leads one to it. This should be followed by which methods were used, if relevant (it usually isn’t) followed by the gist of the results/conclusions. I am aware of the word limit, however if there isn’t space for the hypothesis (which is the whole basis of a scientific study!) in the Abstract, that Abstract hasn’t been written correctly. This Abstract needs to be re-written.
The additional text in the Introduction is useful as this is a way of building a more detailed case to support the study and its context. However, it does not replace the need for a crisp Abstract.
- Analysis of the results
The authors are quite right that it was starch intake rather than (all) carbohydrate, however this is trivial with respect to the need for an improvement in the Discussion, which was recommended by this reviewer. A deeper and more thoughtful analysis is required. I set out several metabolic possibilities in my review, though there are others, and thus the authors claim that the reasons are “unclear” is insufficient, as are the changes that have been made. Specifically:
Lines 268-271 describes a result rather than an analysis of the results. It’s not wrong, but it does not discharge the criticism made.
Lines 326-328: This sentence is a breath-taking and lazy generalisation. The Discussion needs to be much more specific than thisand deeper. As stated before, increased (digestible) carbohydrate intake gives rise to increased de novo lipogenesis which in turn changes which lipids the system is built out of. The well-known relationship of structure and function can then be used as the basis for an explanation of the difference between two groups, perhaps best phrased through an hypothesis of what might be investigated next.
Furthermore, I note that the references used are
- Australian Bureau of Statistics (ABS). Australian Health Survey: Nutrition First Results-Foods and Nutrients, 2011-12. Canberra; Australia ABS; 2014. 567
- Dundar AN, Gocmen D. Effects of autoclaving temperature and storing time on resistant starch formation and its functional and physicochemical properties. Carbohydrate polymers. 2013;97(2):764-71.
- Muir J, O'dea K. Measurement of resistant starch: factors affecting the amount of starch escaping digestion in vitro. The American journal of clinical nutrition. 1992;56(1):123-7.
They are inappropriate references for a discussion of molecular metabolism.
Lines 334-336: Like 268-271, this is not wrong so much as it doesn’t explain anything. The phrase ‘producing metabolites’ is off-centre and vague: small molecules are released and transferred more than ‘produced’ (or do you mean synthesised?) and ‘metabolites’ covers virtually any small molecule in the system. Be clearer about which species you are talking about and what they do. You needn’t list all of them, pick the most important 2-3 or 2-3 classes and talk about those. Have the courage to explain your results.
- Sensitivity analysis
The words ‘sensitivity analysis’ do not appear anywhere in the revised manuscript I received. The Supplementary tables mentioned in the Authors’ reply do not appear in the updated Supplementary information either—and it is not sufficient to bury them here either. The manuscript needs to be revised to incorporate this analysis more clearly. It is disingenuous and unacceptable only to present corrected p-values.
Author Response
Reviewer 2
Taylor et al. Present a revised manuscript ‘Macronutrient intake in pregnancy and child cognitive and behavioural outcomes’. The authors have made some of the revisions, but it is not clear that all of them have been implemented thoroughly enough.
The Abstract.
This has not really been changed—the hypothesis needs to be stated in the Abstract. This can be done in one sentence, after brief indication of the evidence that leads one to it. This should be followed by which methods were used, if relevant (it usually isn’t) followed by the gist of the results/conclusions. I am aware of the word limit, however if there isn’t space for the hypothesis (which is the whole basis of a scientific study!) in the Abstract, that Abstract hasn’t been written correctly. This Abstract needs to be re-written. The additional text in the Introduction is useful as this is a way of building a more detailed case to support the study and its context. However, it does not replace the need for a crisp Abstract.
Author’s response:
Our apologies for omitting to add the hypothesis in the abstract in the previous revision of the manuscript. We have now stated the hypothesis and made the wording of this section more concise, lines 24-44 now reads:
‘Prenatal nutrient exposures can impact on brain development and disease susceptibility across the lifespan. It is well established that maternal macronutrient intake during pregnancy influences foetal and infant development. Therefore, we hypothesise that maternal macronutrient intakes during pregnancy are correlated with cognitive development during early childhood. The current study aimed to investigate the relationship between maternal macronutrient intake during pregnancy and child cognitive and behavioural outcomes at age 4 years. We analysed prospective data from a cohort of 64 Australian mother-child dyads. Maternal macronutrient intake was assessed using a validated 74-item food frequency questionnaire at 2 time-points during pregnancy. Child cognition and behaviour were measured at age years using the validated Wechsler Preschool and Primary Scale of Intelligence, 3rd version (WPPSI-III) and Child Behaviour Checklist (CBC). Linear regression models were used to quantify statistical relationships and were adjusted for maternal age, education, pre-pregnancy BMI, breastfeeding duration and birthweight. Child Performance IQ was inversely associated with maternal starch intake (b=-11.02, p=0.03). However, no other associations were found. Further research is needed to explore the association between different types of starch consumed during pregnancy and child cognitive development.’
Analysis of the results
The authors are quite right that it was starch intake rather than (all) carbohydrate, however this is trivial with respect to the need for an improvement in the Discussion, which was recommended by this reviewer. A deeper and more thoughtful analysis is required. I set out several metabolic possibilities in my review, though there are others, and thus the authors claim that the reasons are “unclear” is insufficient, as are the changes that have been made. Specifically:
Lines 268-271 describes a result rather than an analysis of the results. It’s not wrong, but it does not discharge the criticism made.
Author’s response:
We agree that the sentence at lines 268-271 is a description rather than an interpretation of the results presented. We have now revised this sentence (from the results section), lines 275-278 reads:
‘There was a non-significant trend indicated for each log transformed additional gram of total carbohydrate intake consumed during pregnancy, child Performance IQ decreased (worsened) by approximately 15% (25 points out of a maximum score of 160).’
We have now sought to deepen the discussion section in relation to analysing this result (described above), lines 317-327:
‘This study identified increasing carbohydrate intake during pregnancy was negatively associated with child Performance IQ, a measure of non-verbal reasoning, attention and visuo-spatial processing, however, this relationship was not significant. Although there is evidence to support that an abnormal carbohydrate metabolism during pregnancy such as impaired glucose tolerance (IGT) and gestational diabetes mellitus (GDM) is adversely associated with child cognitive outcomes (62-64). For example, Xu et al., (63) reported that children born to mothers with GDM (n=1421) had lower total wide range assessment of visual motor abilities scores (WRAVMA), a measure of visual-spatial and fine motor ability, at 3 years of age compared with children born to mothers with normal glucose tolerance (n=1187) (-3.09 points; 95% confidence interval (CI) -6.12, -0.05).’
Lines 326-328: This sentence is a breath-taking and lazy generalisation. The Discussion needs to be much more specific than this and deeper. As stated before, increased (digestible) carbohydrate intake gives rise to increased de novo lipogenesis which in turn changes which lipids the system is built out of. The well-known relationship of structure and function can then be used as the basis for an explanation of the difference between two groups, perhaps best phrased through an hypothesis of what might be investigated next.
Furthermore, I note that the references used are
- Australian Bureau of Statistics (ABS). Australian Health Survey: Nutrition First Results-Foods and Nutrients, 2011-12. Canberra; Australia ABS; 2014. 567
- Dundar AN, Gocmen D. Effects of autoclaving temperature and storing time on resistant starch formation and its functional and physicochemical properties. Carbohydrate polymers. 2013;97(2):764-71.
- Muir J, O'dea K. Measurement of resistant starch: factors affecting the amount of starch escaping digestion in vitro. The American journal of clinical nutrition. 1992;56(1):123-7.
They are inappropriate references for a discussion of molecular metabolism.
Author response:
Our apologies for not providing an adequate response to your feedback, we agree that this explanation could be more specific. We have now provided a deeper explanation and provided appropriate molecular metabolism references, lines 341-355 reads:
‘A higher consumption of rapidly digested starches contributes to increased de novo lipogenesis and attenuates the deposition of triglycerides into adipocytes throughout the body (69). Triglycerides are chemically stable when stored in adipocytes (70-72). However, once storage is at saturation, triglycerides may be deposited in non-adipose tissues such as the liver, heart and pancreas, which can lead to lipotoxicity and inflammation (70-72). In the current study, it could be hypothesised that higher intakes of rapidly digested starches alters lipid metabolism and adversely impacts on cell, tissue and organ structure and function in the central nervous system, potentially contributing to a decline in child performance IQ. This could not be evaluated in the current study, as specific types of starch could not be quantified using NUTTAB-95 data. Therefore, further investigation is warranted in future studies where detailed information on types of starch is available.’
References:
- Higgins JA, Brown MA, Storlien LH. Consumption of resistant starch decreases postprandial lipogenesis in white adipose tissue of the rat. Nutrition Journal. 2006;5(1):25.
- Solinas G, Borén J, Dulloo AG. De novo lipogenesis in metabolic homeostasis: More friend than foe? Molecular Metabolism. 2015;4(5):367-77.
- Unger RH, Clark GO, Scherer PE, Orci L. Lipid homeostasis, lipotoxicity and the metabolic syndrome. Biochimica et biophysica acta. 2010;1801(3):209-14.
- Opazo-Ríos L, Mas S, Marín-Royo G, Mezzano S, Gómez-Guerrero C, Moreno JA, et al. Lipotoxicity and Diabetic Nephropathy: Novel Mechanistic Insights and Therapeutic Opportunities. International journal of molecular sciences. 2020;21(7).
Lines 334-336: Like 268-271, this is not wrong so much as it doesn’t explain anything. The phrase ‘producing metabolites’ is off-centre and vague: small molecules are released and transferred more than ‘produced’ (or do you mean synthesised?) and ‘metabolites’ covers virtually any small molecule in the system. Be clearer about which species you are talking about and what they do. You needn’t list all of them, pick the most important 2-3 or 2-3 classes and talk about those. Have the courage to explain your results.
Author’s response:
We have now explained the results in more detail, lines 356-373 reads:
‘Resistant starch is the portion of starch that is resistant to degradation by the enzyme α-amylase in the small intestine (73). Instead, it is fermented in the colon by several bacteria groups (e.g. amylolytic gut bacteria) releasing fermentation products including short-chain fatty acids (acetate, propionate, butyrate, and valerate), branched-chain fatty acids (isovaleric and isobutyric acids), ammonia, amines, phenolic compounds and gases (methane, hydrogen, carbon dioxide) (73). Colonic metabolites including short-chain fatty acids (acetate, propionate, butyrate, and valerate) are associated with a number of health benefits on gastrointestinal health, insulin sensitivity and weight management (74, 75). For example, short-chain fatty acids propionate stimulates the secretion of gut hormone peptides YY (PYY) and glucagon-like peptide 1 (GLP-1) which are essential for appetite regulation and glucose homeostasis (76). Butyrate is associated with being an anti-inflammatory agent by inhibiting the activation of transcription factors, NF-kB, which regulates the expression of genes associated with inflammation (e.g. cytokines, adhesion molecules, acute-phase proteins) (77).’
- Sensitivity analysis
The words ‘sensitivity analysis’ do not appear anywhere in the revised manuscript I received. The Supplementary tables mentioned in the Authors’ reply do not appear in the updated Supplementary information either—and it is not sufficient to bury them here either. The manuscript needs to be revised to incorporate this analysis more clearly. It is disingenuous and unacceptable only to present corrected p-values
Author’s response
We have now described the sensitivity analysis in both the methods and results section, lines 223-225 reads:
‘Sensitivity analyses were conducted using linear regression models that were not adjusted for total energy intake nor confounders to assess the impact on the study results.’
Lines 307-310 reads:
‘Analytic models without adjustment for confounding variables are presented in Tables S3-S4. These sensitivity analyses did not detect any significant differences in results compared to the fully adjusted analyses presented.’
We will double check that the updated supplementary tables uploaded appear in the journal submission system.